# ZerO Initialization: Initializing Neural Networks with only Zeros and Ones

**Jiawei Zhao**                                      *jiawei@caltech.edu*
*California Institute of Technology*

**Florian Schäfer**                          *florian.schaefer@cc.gatech.edu*
*Georgia Institute of Technology*

**Anima Anandkumar**                                 *anima@caltech.edu*
*California Institute of Technology*
*NVIDIA*

**Reviewed on OpenReview:** *https:// openreview. net/ forum? id= 1AxQpKmiTc*

## Abstract

Deep neural networks are usually initialized with random weights, with adequately selected initial variance to ensure stable signal propagation during training. However, selecting the appropriate variance becomes challenging especially as the number of layers grows. In this work, we replace random weight initialization with a *fully deterministic* initialization scheme, viz., ZerO, which initializes the weights of networks with only *zeros and ones* (up to a normalization factor), based on identity and Hadamard transforms. Through both theoretical and empirical studies, we demonstrate that ZerO is able to train networks without damaging their expressivity. Applying ZerO on ResNet achieves state-of-the-art performance on various datasets, including ImageNet, which suggests random weights may be unnecessary for network initialization. In addition, ZerO has many benefits, such as training ultra deep networks (without batch-normalization), exhibiting low-rank learning trajectories that result in low-rank and sparse solutions, and improving training reproducibility[1].

## 1  Introduction

An important question in training deep neural networks is how to initialize the weights. Currently, random weight initialization is the de-facto practice across all architectures and tasks. However, choosing the *variance* of the initial weight distribution is a delicate balance when training deep neural networks. If the variance is too large, it can lead to an excessive amplification of the activations propagating through the network during training, resulting in *exploding gradients*. On the other hand, if the weights are initialized too small, the activations may not propagate at all, resulting in *vanishing gradients*. These issues become more challenging as the number of layers in the network grows.

The above challenges can be avoided if *identity initialization* is used instead. It initializes each layer in the network as an identity mapping, such that the input data can be identically propagated to the network output. In this case, there is no need to introduce any randomness or consider its variance. Identity initialization is well studied theoretically from an optimization perspective. Hardt & Ma (2017) prove the existence of a global minimum close to the identity parameterization in a deep residual network. Bartlett et al. (2019) further prove the rate of convergence of gradient-based optimization under identity initialization.

---

[1]Code repository: `https://github.com/jiaweizzhao/ZerO-initialization`.

Table 1: Comparing ZerO initialization with randomly perturbed identity initialization. $F$ represents a transformation[1] of an arbitrary layer $l$.

| Settings | Techniques | Related Works |
|---|---|---|
| $\boldsymbol{x}_{l+1} = \boldsymbol{x}_l + F(\boldsymbol{x}_l)$ | $F$ is randomly initialized with small variance | Hardt & Ma (2017) |
| $\boldsymbol{x}_{l+1} = \boldsymbol{x}_l + Conv(F(\boldsymbol{x}_l))$ | $F$ is randomly initialized, kernel in $Conv$ is zero | Zhang et al. (2019) |
| $\boldsymbol{x}_{l+1} = \boldsymbol{x}_l + Norm(F(\boldsymbol{x}_l))$ | $F$ is randomly initialized, scale in $Norm$ is zero | Goyal et al. (2017) |
| $\boldsymbol{x}_{l+1} = \boldsymbol{x}_l + \alpha(F(\boldsymbol{x}_l))$ | $F$ is randomly initialized, scalar $\alpha$ is zero | Bachlechner et al. (2020) |
| **All settings above** | **$F$ is a Hadamard/identity transform[2]** | **ZerO (ours)** |

However, prior theoretical works on identity initialization assume that all layers had the same dimensionality, which does not hold for practical networks. Typically, practical networks have *varying dimensionality* across layers, such as the variations of spatial and channel dimensions in ResNet architectures (He et al., 2016). Directly applying identity initialization to these networks leads to a problem of *training degeneracy*, as our theoretical study will demonstrate later.

To avoid the training degeneracy, previous works (summarized in Table 1) employ identity initialization with *random perturbations* to facilitate escapes from a saddle point or to break feature symmetry (Blumenfeld et al., 2020). Broadly, these approaches satisfy the property of *dynamical isometry*, to preserve the signal propagation and ensure well-behaved gradients at initialization (Saxe et al., 2014). Despite the efficacy of random perturbations, they inevitably introduce additional tuning of variances, which can result in gradient explosion in deep networks without careful tuning.

**Our work:** we propose ZerO initialization that removes *all randomness in the weight initialization*. As illustrated in Figure 1, ZerO initializes networks with Hadamard and identity transforms, which assigns all the weights to only *zeros and ones*.

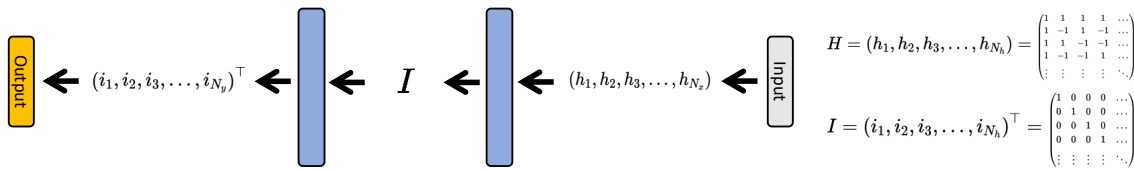

Figure 1: Illustrating ZerO in a 3-layer network with input dimension $N_x$, hidden dimension $N_h$, and output dimension $N_y$, where $N_h > N_x, N_y$. $\boldsymbol{H}$ and $\boldsymbol{I}$ are $N_h \times N_h$ Hadamard and identity matrix, respectively. The dimension-increasing layer is initialized by columns of the Hadamard matrix. The rest layers are initialized by identity matrix or rows of it.

ZerO is not affected by the problem of training degeneracy or accuracy loss. Compared to random initialization, ZerO provides state-of-the-art performance over various image classification tasks, including ImageNet. We further discover many unique properties and benefits of ZerO:

**Stable training without batch normalization.** ZerO ensures well-behaved signal propagation, which provides stable training without batch normalization. Testing ResNet with over 500 layers, we find ZerO converges faster than carefully designed random methods, such as Fixup and ReZero (Zhang et al., 2019; Bachlechner et al., 2020).

**Low-rank learning trajectory.** We find that ZerO exhibits a low-rank learning trajectory, where the rank of each matrix gradually increases during training. We believe this is the first time that the greedy low-rank

---

[1]The transformation may contain linear, nonlinear or convolutional operations.
[2]Both identity and Hadamard transforms are deterministic parameterizations.

learning (GLRL) trajectory, a theoretical characterization of gradient descent, has been observed in large-scale deep learning applications. GLRL is a consequence of implicit rank regularization by gradient descent under infinitesimal initialization (Li et al., 2021; Razin et al., 2021). It can be viewed as performing a rank-constrained optimization and greedily relaxing the rank restriction by one whenever it fails to reach a global minimizer. GLRL has been used to explain the excellent generalization in gradient-based deep learning, as it converges to a global (or local) minima with the minimum rank. However, the GLRL trajectory has never been observed in practice due to its impractical requirement of infinitesimal initialization.

**Sparse and low-rank solutions.** We observe that ZerO-initialized networks converge to sparse and low-rank solutions. Compared to randomly initialized networks, the sub-networks obtained in trained ZerO-initialized networks achieve 30% lower (matrix or tensor) rank or 25% higher sparsity without sacrificing accuracy.

**Better training reproducibility.** Since ZerO does not require any random perturbations, it is a fully deterministic initialization scheme. Unlike determinism in random initialization, which needs fixing pseudorandom number generators in hardware, the weights initialized by ZerO are fixed regardless of how the random seed varies or which hardware is used. ZerO significantly reduces the training variation and thus achieves better training reproducibility (the remaining randomness is only due to batch selection). Compared to random initialization, ZerO produces 20%-40% lower standard deviation of the final accuracy over repeated experiments with different random seeds.

**Theoretical analysis of ZerO.** Theoretically, we demonstrate that ZerO breaks a training degeneracy that arises when applying identity initialization to networks with varying dimensionality across layers. We prove that the training degeneracy necessarily occurs in standard identity initialization because the rank of any $N_h \times N_h$ matrix in the network is upper bounded by the input and output dimensions $N_x$ and $N_y$ throughout the entire training, no matter how large the size of $N_h$ is. This limits the expressivity of each matrix, resulting in the degeneracy of training.

**Our contributions are summarized as follows:**

1. We design ZerO initialization, the first fully deterministic initialization that achieves state-of-the-art performance in practice.

2. ZerO is backed with theoretical understanding. As shown in Theorem 1, we prove how ZerO breaks the training degeneracy by applying Hadamard transforms.

3. ZerO has many benefits, such as training ultra deep networks (without batch-normalization), exhibiting low-rank learning trajectory, converging to sparse and low-rank solutions, and improving training reproducibility.

## 2 Is Randomness Necessary in Identity Initialization?

### 2.1 Background

We wish to train a function $\mathcal{F}(\boldsymbol{x})$ to learn a particular input-output map given a set of $P$ training samples $(\boldsymbol{x}^\mu, \boldsymbol{y}^\mu) \in \mathbb{R}^{N_x \times N_y}$, where $\mu = 1, ..., P$. Training is accomplished by minimizing the squared error $\mathcal{L} = \frac{1}{2} \sum_{\mu=1}^{P} \|\boldsymbol{y}^\mu - \mathcal{F}(\boldsymbol{x}^\mu)\|_2^2$ using gradient descent with a step size $\eta$.

We model $\mathcal{F}(\boldsymbol{x})$ to be a multilayer perceptron with $L > 2$ hidden layers, such that:

$$\boldsymbol{x}_l = \boldsymbol{W}_l \boldsymbol{z}_{l-1} \qquad \boldsymbol{z}_l = \varphi(\boldsymbol{x}_l),$$

with $l \in 1, ..., L$. Let $\boldsymbol{z}_0 = \boldsymbol{x}$ and $\mathcal{F}(\boldsymbol{x}) = \boldsymbol{z}_L$. $\varphi$ is an element-wise nonlinearity. We assume that $\mathcal{F}$ has uniform hidden dimension $N_h$, with $\boldsymbol{W}_l \in \mathbb{R}^{N_h \times N_h}$ for $l \in 2, ..., L-1$, $\boldsymbol{W}_1 \in \mathbb{R}^{N_h \times N_x}$, and $\boldsymbol{W}_L \in \mathbb{R}^{N_y \times N_h}$.

The input-output Jacobian is a well-studied proxy for estimating the stability of signal propagation at initialization (Saxe et al., 2014). It is defined as:

$$\boldsymbol{J}_{io} = \frac{\partial \boldsymbol{z}_L}{\partial \boldsymbol{z}_0}.$$

Proposed by (Saxe et al., 2014), dynamical isometry is a condition where all singular values of the Jacobian $\boldsymbol{J}_{io}$ concentrate near 1. If $\boldsymbol{J}_{io}$ is well-conditioned, the backpropagation error $\frac{\partial \mathcal{L}}{\partial \boldsymbol{z}_l}$ at any layer $l$ will be well-conditioned as well. This ensures stable signal propagation and well-behaved gradients at initialization.

Consider the case of $N_x = N_y = N_h$. Identity initialization is defined as initializing each matrix to be an identity matrix: $\boldsymbol{W}_l = \boldsymbol{I}$. In this case, the dynamical isometry property for a linear $\mathcal{F}$ can be easily verified as $\boldsymbol{J}_{io} = \boldsymbol{I}$. It also holds for certain nonlinearities when applying the identity initialization on residual networks: $\boldsymbol{z}_l = \varphi(\boldsymbol{x}_l) + \boldsymbol{x}_{l-1}$ where $\boldsymbol{W}_l = 0$, such that no signal is passed through the residual branches at initialization. Table 1 lists a few related examples.

From an optimization perspective, Hardt & Ma (2017) suggest that $\mathcal{F}$ has a global minimum very close to its identity initialization, such that $\max_{1 \le l \le L} \|\boldsymbol{W}_l'\| \le O(1/L)$ for large $L$, where $\boldsymbol{W}' = \boldsymbol{W} - \boldsymbol{I}$. Bartlett et al. (2019) also proves that under the identity initialization, gradient descent learns an $\epsilon$-approximation of $\mathcal{F}$ within iterations polynomial in $log(1/\epsilon)$.

## 2.2 Extending to Large Hidden Dimension

So far, we have discussed identity initialization in the special case of fixed dimensionality. Now we extend our discussion to a more practical setting where $\mathcal{F}$ is equipped with a large hidden dimension, such that $N_h >> N_x, N_y$. We also focus on the specific case where $\varphi$ is a Relu nonlinearity.

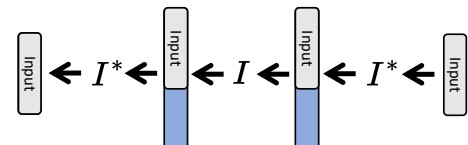

Figure 2: A 3-layer network $\mathcal{F}$ ($N_h > N_x = N_y$) where $\boldsymbol{W}_1, \boldsymbol{W}_3 = \boldsymbol{I}^*$ and $\boldsymbol{W}_2 = \boldsymbol{I}$ at initialization.

A straightforward approach to identity initialization in this case is to initialize $\boldsymbol{W}_1$ such that it projects the input into a $N_x$-dimensional subspace of the $N_h$-dimensional hidden space. This can be achieved by initializing $\boldsymbol{W}_1$ with a *partial identity matrix*:

**Definition 1** (Partial Identity Matrix). Let $\boldsymbol{I}^* \in \mathbb{R}^{l \times r}$, the partial identity matrix $\boldsymbol{I}^*$ is defined as follows:

$$\boldsymbol{I}^* = \begin{cases} (\boldsymbol{I}, \boldsymbol{0}) & \text{where } \boldsymbol{I} \in \mathbb{R}^{l,l} \text{ and } \boldsymbol{0} \in \mathbb{R}^{l,r-l} & \text{if } l < r, \\ \boldsymbol{I} & \text{where } \boldsymbol{I} \in \mathbb{R}^{l,l} & \text{if } l = r, \\ (\boldsymbol{I}, \boldsymbol{0})^\top & \text{where } \boldsymbol{I} \in \mathbb{R}^{r,r} \text{ and } \boldsymbol{0} \in \mathbb{R}^{r,l-r} & \text{otherwise.} \end{cases}$$

For a vector $\boldsymbol{a} \in \mathbb{R}^r$, if $l < r$, then $\boldsymbol{I}^*(\boldsymbol{a})$ clips the last few dimension such that $\boldsymbol{I}^*(\boldsymbol{a}) = (a_1, a_2, ..., a_l)^\top$. If $l > r$, then $\boldsymbol{I}^*$ pads $l - r$ additional dimensions with zero, such that $(a_1, a_2, ..., a_r, 0...0)^\top$. This is also known as a zero-padding operator, such as used in channel-expanding layers in ResNet (He et al., 2016). In the network $\mathcal{F}$, $\boldsymbol{I}^*$ only needs to be applied in the dimension-varying matrices $\boldsymbol{W}_1$ and $\boldsymbol{W}_L$, while leaving the remaining $N_h \times N_h$ matrices to be identity matrix $\boldsymbol{I}$.

We visualize this process in Figure 2. This may seem like a natural extension of identity initialization to a large width setting, but we will show in Section 2.3 it suffers from a problem we call "training degeneracy". To avoid the problem, we use the Hadamard matrix $\boldsymbol{H}$ to initialize the dimension-increasing matrices, such that $\boldsymbol{W}_1 = \boldsymbol{H}\boldsymbol{I}^*$. A Hadamard matrix is defined as follows:

**Definition 2** (Hadamard matrix). For any Hadamard matrix $\boldsymbol{H} = \boldsymbol{H}_m \in \mathbb{R}^{2^m \times 2^m}$ where $m$ is a positive integer, we define $\boldsymbol{H}_0 = 1$ by the identity, and the matrix with large $m$ is defined recursively:

$$\boldsymbol{H}_m = \begin{pmatrix} \boldsymbol{H}_{m-1} & \boldsymbol{H}_{m-1} \\ \boldsymbol{H}_{m-1} & -\boldsymbol{H}_{m-1} \end{pmatrix} = \begin{pmatrix} 1 & 1 & 1 & 1 & \cdots \\ 1 & -1 & 1 & -1 & \cdots \\ 1 & 1 & -1 & -1 & \cdots \\ 1 & -1 & -1 & 1 & \cdots \\ \vdots & \vdots & \vdots & \vdots & \ddots \end{pmatrix} \in \mathbb{R}^{2^m \times 2^m}.$$

The linear transformation described by the Hadamard matrix, called the Hadamard transform, rotates the coordinate axes to be equally weakly aligned with the standard basis. For example, in a two-dimensional plane, the Hadamard transform rotates the standard basis by an exact angle of 45 degree. This turns out to be an important property for breaking the training degeneracy.

### 2.3 Identity Initialization limits Network Expressivity

We now present our main result differentiating the training behavior of different initialization methods and describing the problem of training degeneracy.

**Theorem 1.** *Let $\mathcal{F}$ be a neural network with $L$ matrices, where $\boldsymbol{W}_l \in \mathbb{R}^{N_h \times N_h}$ for $l \in 2, ..., L-1$, $\boldsymbol{W}_1 \in \mathbb{R}^{N_h \times N_x}$, and $\boldsymbol{W}_L \in \mathbb{R}^{N_y \times N_h}$. $\mathcal{F}$ has a uniform hidden dimension $N_h$, input dimension $N_x$, and output dimension $N_y$, where $N_h \geq N_x, N_y$. Define residual component $W_l' = W_l - \boldsymbol{I}$. Let $\boldsymbol{z}_l(\boldsymbol{x})$ to be the activation in the $l$-th layer under the input $\boldsymbol{x}$. Then we have the following results for different initializations:*

(i) *Consider a random perturbation $\mu \in \mathbb{R}^{\mathbb{N}_h \times \mathbb{N}_h}$ where each element is sampled from a Gaussian distribution: $\mu_{ij} \sim \mathcal{N}(0, \sigma^2)$. It is well-known that the randomly perturbed matrices $\boldsymbol{W}_l = \boldsymbol{I} + \mu_l$ (for $l \neq 1, L$) are full-rank almost surely:*

$$Prob(\text{rank}(\boldsymbol{W}_l') = N_h) = 1 \quad for\ l \in 2, ..., L-1. \tag{1}$$

(ii) *When initializing $\boldsymbol{W}_1, \boldsymbol{W}_L = \boldsymbol{I}^*$ and the remaining matrices as $\boldsymbol{W}_l = \boldsymbol{I}$ (for $l \neq 1, L$), for any $\boldsymbol{x} \in \mathbb{R}^{N_x}$, the linear dimension of the set of all possible activations is bounded throughout training as*

$$\dim\left(\text{span}\left(\left\{\boldsymbol{z}_l(\boldsymbol{x}) \middle| \boldsymbol{x} \in \mathbb{R}^{N_x}\right\}\right)\right) \leq N_x \quad for\ l \in 2, ..., L-1. \tag{2}$$

*As a result, the ranks of the weight matrices remain bounded throughout training as*

$$\text{rank}(\boldsymbol{W}_l') \leq N_x \quad for\ l \in 2, ..., L-1. \tag{3}$$

(iii) *When initializing $\boldsymbol{W}_1 = \boldsymbol{H}\boldsymbol{I}^*$, $\boldsymbol{W}_L = \boldsymbol{I}^*$, and the remaining matrices as $\boldsymbol{W}_l = \boldsymbol{I}$ (for $l \neq 1, L$) it is possible for the activations at an intermediate layer to attain*

$$\dim\left(\text{span}\left(\left\{\boldsymbol{z}_l(\boldsymbol{x}) \middle| \boldsymbol{x} \in \mathbb{R}^{N_x}\right\}\right)\right) > N_x, \tag{4}$$

*breaking the constraint on the linear dimension described in Equation 2.*

**Remark 1.** *The constraints on linear dimensions and matrix ranks in Equation 2 and 3 suggest that no matter how large hidden dimension $N_h$ is, the network $\mathcal{F}$ is only optimized within a low-dimensional subspace (depending on input dimension $N_x$) of the full parameter space. This restricts the maximum network expressivity of $\mathcal{F}$, and thus the training may only converge to an underfitting regime, leading to **training degeneracy**.*

**Remark 2.** *Under the assumptions of* (iii)*, the breaking of training degeneracy described in Equation 4 appears to be the generic case. As verified empirically in Figure 3, applying the Hadamard transform in* (ii) *also breaks the rank constraint in Equation 3.*

**Remark 3.** (i) *and* (iii) *avoid the training degeneracy from different directions. Unlike* (i)*, the proposed* (iii) *doesn't introduce any randomness with the help of the Hadamard transform.*

Almost all existing works use random weight initialization, which largely affects the rank of each matrix as shown in (i). (i) can be proved by showing any column (or row) in a random matrix is linearly independent to the other columns (or rows), almost surely. A detailed proof can be found in the appendix.

Consider identity initialization without any randomness. In (ii), $\boldsymbol{W}_1$ identically maps the input $\boldsymbol{x}$ into a subspace of the hidden layer $\boldsymbol{z}_1$, such that $\boldsymbol{z}_1 = (\boldsymbol{x}^\top, 0, ..., 0)^\top$. Thus, the linear dimension on $\boldsymbol{z}_l$ (i.e., the linear dimension on activations $\boldsymbol{z}_l^1, ..., \boldsymbol{z}_l^P$) is equivalent to the linear dimension on $\boldsymbol{x}$, which is upper bounded by $N_x$. This result is held for every layer $\boldsymbol{z}_l$ (where $l \neq 1, L$).

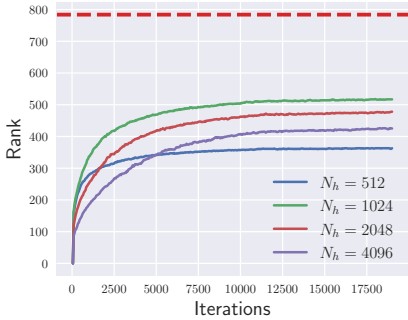 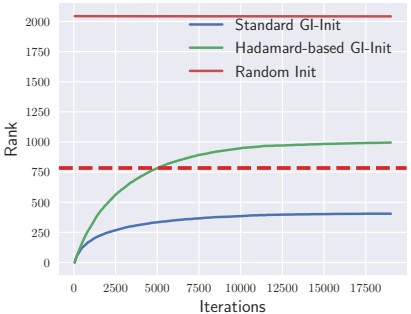

Figure 3: *Verify Theorem 1 in practice. We train a network $\mathcal{F}$ with $L = 3$ on MNIST, and visualize $\mathrm{rank}(W_2)$ over the training. The red dash line denotes the rank constraint on $W_2$ predicted by the theorem, which is $N_x = 784$. **Left:** we verify* (ii) *in the theorem by varying $N_h$. No matter how large $N_h$ is, $\mathrm{rank}(W_2)$ follows the rank constraint through the entire training. **Right:** we verify* (iii) *where applying Hadamard transform breaks the rank constraint introduced in* (ii) *, given $N_h = 2048$. We denote the initializations in* (ii) *and* (iii) *as standard and Hadamard-based GI-Init, respectively. As predicted in* (i)*, random initialization achieves its maximum rank immediately after the initialization.*

To show the rank constraint in Equation 3, we track a single derivative of the residual component at layer $l$:

$$\frac{\partial \mathcal{L}}{\partial \boldsymbol{W}_l'} = \sum_{\mu=1}^{P} \frac{\partial \mathcal{L}}{\partial \boldsymbol{x}_l^\mu} \otimes \boldsymbol{z}_{l-1}^\mu, \tag{5}$$

where $\otimes$ denotes the outer product. We use the following well-known fact:

**Lemma 1.** *Consider a matrix $\boldsymbol{M}$ to be a sum of vector outer products:$\boldsymbol{M} = \sum_{\mu=1}^{Q} \boldsymbol{a}^\mu \otimes \boldsymbol{b}^\mu$, where $\boldsymbol{a}^\mu \in \mathbb{R}^{N_a}$ and $\boldsymbol{b}^\mu \in \mathbb{R}^{N_b}$ for $\mu \in 1, ..., Q$. Let $Q > N_a, N_b$. If linear dimensions $\dim(\mathrm{span}(\boldsymbol{a}^1, ..., \boldsymbol{a}^Q)) \leq U$ and $\dim(\mathrm{span}(\boldsymbol{b}^1, ..., \boldsymbol{b}^Q)) \leq V$, where $U \leq N_a$ and $V \leq N_b$, then:*

$$\mathrm{rank}(\boldsymbol{W}) \leq \min(U, V)$$

.

By Lemma 1, at initialization, the upper bound $N_x$ on the linear dimension on $\boldsymbol{z}_{l-1}$ results in a rank constraint on $\mathrm{rank}(\boldsymbol{W}_l')$. The rank constraint holds during the entire training as $\frac{\partial \mathcal{L}}{\partial \boldsymbol{W}_l'}$ has a zero-valued $N_y \times (N_h - N_x)$ sub-matrix at every iteration (as shown in the appendix). Since $W_l' = \sum_{t=1}^{T} -\eta \frac{\partial \mathcal{L}}{\partial \boldsymbol{W}_l'}\big|_t$ after $T$ weight updates (by gradient descent with a step size $\eta$), $\mathrm{rank}(W_l')$ is bounded by $N_x$ no matter what $T$ is. This results in the training degeneracy as described in Remark 1. We also verify it empirically in Figure 3.

To avoid the training degeneracy, we need to overcome limitations on the linear dimension of the set of possible activations. This is indeed possible when using the Hadamard matrix as $\boldsymbol{W}_1 = \boldsymbol{H}\boldsymbol{I}^*$, as we will illustrated by means of an example.

**Lemma 2.** *Assume $N_h = 4$ and $N_x = 3$. For any vector $\boldsymbol{x} \in \mathrm{span}(\boldsymbol{e}_2, \boldsymbol{e}_3)$ where $\boldsymbol{e}_2$ and $\boldsymbol{e}_3$ are coordinate vectors $(0, 1, 0)^\top$ and $(0, 0, 1)^\top$, it holds that:*

$$\mathrm{span}\left(\{\boldsymbol{z}_1(\boldsymbol{x})|\boldsymbol{x} \in \mathbb{R}^{N_x}\}\right) = \mathrm{span}(\mathrm{Relu}(\boldsymbol{H}\boldsymbol{I}^*\boldsymbol{e}_2), \mathrm{Relu}(-\boldsymbol{H}\boldsymbol{I}^*\boldsymbol{e}_2), \mathrm{Relu}(\boldsymbol{H}\boldsymbol{I}^*\boldsymbol{e}_3), \mathrm{Relu}(-\boldsymbol{H}\boldsymbol{I}^*\boldsymbol{e}_3)), \tag{6}$$

*where $\mathrm{Relu}(\boldsymbol{H}\boldsymbol{I}^*\boldsymbol{e}_2)$, $\mathrm{Relu}(-\boldsymbol{H}\boldsymbol{I}^*\boldsymbol{e}_2)$, $\mathrm{Relu}(\boldsymbol{H}\boldsymbol{I}^*\boldsymbol{e}_3)$, and $\mathrm{Relu}(-\boldsymbol{H}\boldsymbol{I}^*\boldsymbol{e}_3)$ are linearly independent. This indicates that:*

$$\dim\left(\mathrm{span}\left(\{\boldsymbol{z}_1(\boldsymbol{x})|\boldsymbol{x} \in \mathbb{R}^{N_x}\}\right)\right) = 4 = N_h > N_x = 3.$$

When using Hadamard matrix as $\boldsymbol{W}_1 = \boldsymbol{H}\boldsymbol{I}^*$, the breaking of the training degeneracy described in Lemma 2 appears to be the generic case. As verified empirically in Figure 3, this also breaks the rank constraint in Equation 3.

We point out that the increase of the linear dimension of the set of possible $z_l$ is only possible due to the nonlinearity. If $\mathcal{F}$ is a linear network, the linear dimension on every $z_l$ is at most $N_x$, no matter how the weights are initialized.

Nevertheless, the nonlinearity can not increase the linear dimensionality if we initialize the network with a partial identity matrix. This is because when $W_1 = I^*$, span $\left(\left\{z_l(x)\middle|x \in \mathbb{R}^{N_x}\right\}\right)$ is aligned with the standard basis, i.e., each vector in the span at least has $N_h - N_x$ zero coefficients when expressed in the standard basis. Thus, an element-wise nonlinearity can not increase the linear dimension of its input beyond $N_x$.

To break the alignment span $\left(\left\{z_l(x)\middle|x \in \mathbb{R}^{N_x}\right\}\right)$ with the standard basis, we use the Hadamard transform. This is because it transforms the subspace such that the new basis is equally weakly aligned with the standard basis. We note that other linear transforms may also detach the subspace from the standard basis, but the Hadamard transform is the most natural choice.

## 3 ZerO Initialization

The initialization analyzed in (iii) of Theorem 1 is based on a network condition in which all hidden spaces $z_1, ..., z_{L-1}$ have the same dimension $N_h$. Motivated by our theoretical understanding, we propose ZerO initialization, which initializes the weights of any network with arbitrary hidden dimensions. As described in Algorithm 1, ZerO only initializes dimensional-increasing layers with Hadamard matrices to avoid the training degeneracy. Other layers are simply initialized by (partial) identity matrices. We also rescale Hadamard matrices by a normalization factor $2^{-(m-1)/2}$, resulting in an orthonormal Hadamard transform.

---

**Algorithm 1** *ZerO Initialization.*

---

**Input:** a neural network $\mathcal{F}$ with $L$ matrices $W_l \in \mathbb{R}^{P_l \times Q_l}$ for $l$ in $1, ..., L$. $I^*$ is partial identity matrix defined in Definition 1. $H_m$ is the Hadamard matrix defined in Definition 2.
**For** $l$ **in** $1, ..., L$**:**
    **If** $P_l = Q_l$**:** $W_l \leftarrow I$                                               ▷ Identity mapping
    **If** $P_l < Q_l$**:** $W_l \leftarrow I^*$                                   ▷ Propagate the first $P_l$ dimensions
    **If** $P_l > Q_l$**:** $W_l \leftarrow c\, I^* H_m I^*$, where $m = \lceil \log_2(P_l) \rceil$ and $c = 2^{-(m-1)/2}$     ▷ Apply Hadamard matrix

---

We also apply ZerO to the well-developed ResNet architectures in He et al. (2016). As shown in Algorithm 2, we apply ZerO to convolution in a similar way by considering the variation in channel dimensions. When $K$ is a 1x1 convolution, $K$ also can be viewed a $c_{out} \times c_{in}$ matrix, which matches the initialization in Algorithm 1. We note that Algorithm 2 can be applied to every convolution in ResNet, including the first 3x3 convolution, 1x1 convolutions in spatial-downsampling skip connections, and convolutions in basic block and bottleneck block.

To achieve dynamical isometry at initialization, we apply a common technique that initializes the last convolution in each residual block as zero. This helps suppress the signals from residual branches to stabilize signal propagations, as studied in Zhang et al. (2019); Bachlechner et al. (2020).

---

**Algorithm 2** *ZerO Initialization on Convolution.*

---

**Input:** number of input channels $c_{in}$, number of output channels $c_{out}$, odd kernel size $k$.
**Return:** a $c_{out} \times c_{in} \times k \times k$ convolutional kernel $K$.
**Let** $n \leftarrow \lfloor k/2 \rfloor$
    **If** $c_{out} = c_{in}$**:** $K[:, :, n, n] \leftarrow I$
    **If** $c_{out} < c_{in}$**:** $K[:, :, n, n] \leftarrow I^*$
    **If** $c_{out} > c_{in}$**:** $K[:, :, n, n] \leftarrow c\, I^* H_m I^*$, where $m = \lceil \log_2(P_l) \rceil$ and $c = 2^{-(m-1)/2}$

---

We also apply ZerO to networks with or without batch normalization. For ResNet with batch normalization, we follow the standard practice to initialize the scale and bias in batch normalization as one and zero,

| Dataset | Model | Initialization | Test Error (mean $\pm$ std) |
|---|---|---|---|
| CIFAR-10 | ResNet-18 | **ZerO Init** | $5.13 \pm 0.08$ |
| | | Kaiming Init | $5.15 \pm 0.13$ |
| | | Xavier Init | $5.23 \pm 0.16$ |
| ImageNet | ResNet-50 | **ZerO Init** | $23.43 \pm 0.04$ |
| | | Kaiming Init | $23.46 \pm 0.07$ |
| | | Xavier Init | $23.65 \pm 0.11$ |

Table 2: Benchmarking ZerO on CIFAR-10 and ImageNet. We repeat each run 10 times with different random seeds.

respectively. For training without batch normalization, we adopt a technique proposed by Zhang et al. (2019), where the batch normalization is replaced by learnable scalar multipliers and biases.

## 4 Experiments

In this section, we empirically benchmark ZerO on CIFAR-10 and ImageNet datasets, where we evaluate ResNet-18 on CIFAR-10 and ResNet-50 on ImageNet (Krizhevsky, 2009; Deng et al., 2009). Both ResNet structures follow the design from He et al. (2016), which includes batch normalization by default.

**Hyperparameter settings.** We find that *ZerO can fully utilize the default hyperparameters*, which include a learning rate of 0.1, a momentum of 0.9, and a weight decay of 0.0001. In addition, we observe the learning rate warmup is essential for ZerO to achieve a large maximal learning rate, as most of the weights start from the exact zero. We warm up the learning rate with 5 and 10 epochs for ImageNet and CIFAR-10, respectively.

We present our main results that compare different initialization schemes. For each dataset, all experiments use the same hyperparameter settings by default. Each experiment is repeated for ten runs with different random seeds. As shown in Table 2, ZerO achieves state-of-the-art accuracy on both datasets compared to other random methods.

In addition, we compare ZerO with a broad range of related works on CIFAR-10 using ResNet-18 and ResNet-50, including ReZerO (Bachlechner et al., 2020), Fixup (Zhang et al., 2019), SkipInit (De & Smith, 2020) and ConstNet (Blumenfeld et al., 2020). As shown in Table 3, ZerO consistently achieves top performance compared to other methods.

We note that the ConstNet proposed by Blumenfeld et al. (2020) is also a deterministic initialization. However, unlike ZerO which preserves feature diversity, ConstNet is designed to eliminate the diversity by averaging the features through layers. The feature symmetric problem in ConstNet causes significant degradation, and additional random noise (e.g., non-deterministic GPU operation and dropout) is needed to break the symmetry.

| Method | ZerO | ReZero | Fixup | SkipInit | ConstNet | ConstNet* |
|---|---|---|---|---|---|---|
| **ResNet-18** | 5.13 | 5.20 | 5.17 | 5.26 | 72.39 | 5.41 |
| **ResNet-50** | 4.53 | 4.72 | 4.51 | 4.63 | 71.58 | 4.88 |

Table 3: Compare ZerO with other initialization methods on CIFAR-10. ConstNet* denotes ConstNet with non-deterministic GPU operations discussed in Blumenfeld et al. (2020). Top-1 test error is reported.

**Training ultra deep network without batch normalization** Although there are methods attempting to train networks without batch normalization (by achieving dynamical isometry), they inevitably introduce random perturbations at initialization, affecting the training stability when the network is sufficiently deep (Zhang et al., 2019; De & Smith, 2020). We benchmark ZerO with state-of-the-art methods on training without batch normalization. As shown in Figure 4 (left), compared to other methods, ZerO achieves the best training stability for networks with even around 500 layers. It also matches the baseline where batch normalization is enabled.

**Improved reproducibility.** In addition, as shown in Table 2, ZerO achieves the lowest standard deviation over the repeated runs. On ImageNet, the gap between ZerO and other methods is even more than 40%. Thus, removing the randomness in the weight initialization improves reproducibility, with possible implications for topics such as trustworthy machine learning and network interpretation.

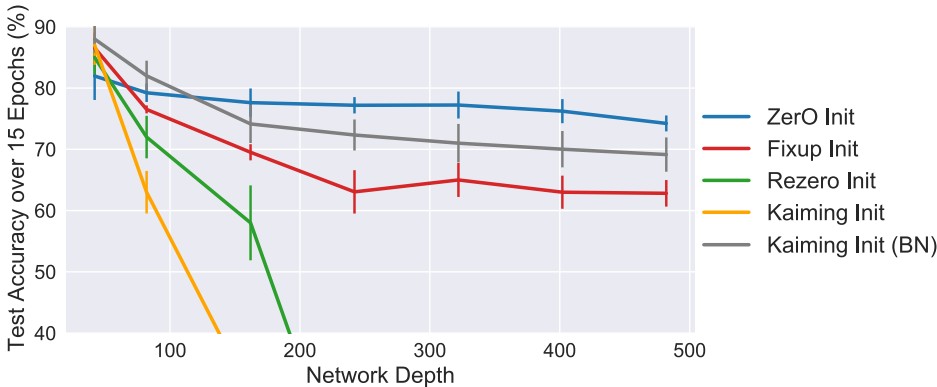

Figure 4: Training extreme deep ResNet on CIFAR-10 over 15 epochs.

### 4.1 ZerO on Transformer

We also apply ZerO to Transformer and evaluate it on WikiText-2 dataset (Vaswani et al., 2017). In each Transformer layer, we use ZerO to initialize both multi-head attention and feed-forward layers. Because the embedding size is fixed in the multi-head attention, we initialize the projection matrix of queries $W_Q$ as identity and the projection matrices of keys and values $W_K, W_V$ at zero. For the feed-forward layers, we initialize the connection matrices according to their hidden dimensions using Algorithm 1.

We train the Transformer models for 20 epochs with a single learning rate decay at epoch 10 [2]. We also vary the number of layers in the model from 2 to 20. As shown in Table 4, ZerO achieves similar performance compared to the standard initialization. In addition, it has better training stability over deeper Transformer models, which is consistent with our previous results on ResNet.

| Number of layers | 2 | 4 | 6 | 8 | 10 | 20 |
|---|---|---|---|---|---|---|
| Standard | 200.44 | 168.69 | 154.67 | 146.43 | diverged | diverged |
| ZerO | 192.34 | 169.73 | 151.91 | 149.27 | 145.62 | 141.81 |

Table 4: Evaluate Transformer on WikiText-2. We vary the number of layers in Transformer, where each layer consists of a multi-head attention and a feed-forward layer. Test perplexity is reported (lower is better).

---

[2]We use a transformer architecture (provided by the link here) that was smaller than the transformers typically used for this task, explaining the general degradation of the results.

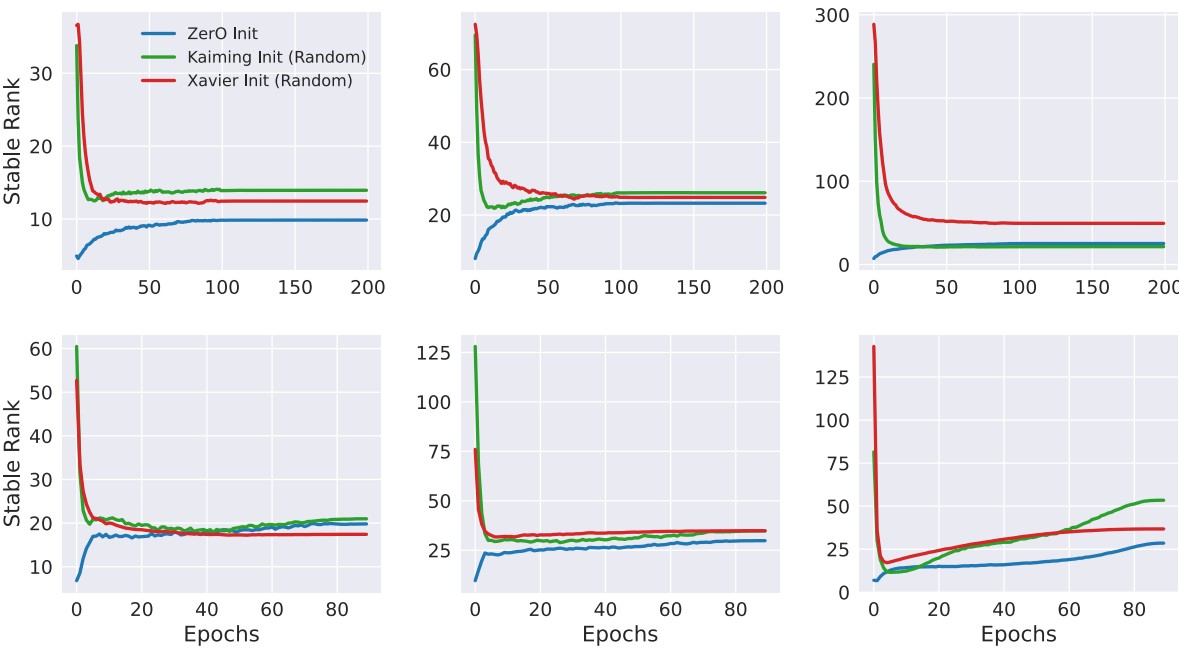

Figure 5: *Low-rank training trajectories in ResNet-18 on CIFAR-10 (top row) and ResNet-50 on ImageNet (bottom row). We visualize trajectories of the first convolutions in second, third, and fourth groups of residual blocks in ResNet.*

## 5 Low-Rank Learning Trajectory

Although ZerO and random initialization achieve similar test accuracies, their training trajectories differ significantly. In contrast to random initialization, which begins optimization from a complex network (i.e., full-rank weight matrices, as shown in Figure 3), ZerO starts the training from a "simple" network and gradually increases its complexity.

To show the difference in practice, we track the ranks of convolutional kernels in ResNets during training, where the rank of each kernel can reflect its complexity. We measure the stable rank, which is defined as

$$\|\boldsymbol{W}\|_F^2 / \|\boldsymbol{W}\|_2^2 = \sum_{i=1}^{k} \sigma_i^2(\boldsymbol{W})/\sigma_{max}^2(\boldsymbol{W}),$$

for any matrix $\boldsymbol{W}$ with k singular values $\sigma_i$. The stable rank is a soft version of the operator rank, and unlike the operator rank, it is insensitive to small singular values. We compare the stable ranks of various kernels between ZerO and random initialization during training. As shown in Figure 5, in contrast to random methods that begin with extremely high stable ranks, ZerO starts with low stable ranks and gradually increases them during training.

We believe ZerO's learning trajectory is the first demonstration of greedy low-rank learning (GLRL) in large-scale deep learning applications. GLRL is a theoretical characterization of gradient descent, such that: when matrices are initialized with infinitesimal values, gradient descent performs a rank-constrained optimization and greedily relaxes the rank restriction by one whenever it fails to reach a minimizer (Li et al., 2021; Razin et al., 2021).

For example, when a matrix is initialized sufficiently small (where its rank is approximately zero), gradient descent first searches the solution over all rank-one matrices. If it fails to find a minimizer, it will relax the rank constraint by one and search again over all rank-two matrices. The search is stopped at rank-$n$ if it finds a minimizer among all rank-$n$ matrices.

GLRL suggests that gradient descent implicitly biases the model towards simple solutions by searching through the solution space in an incremental order of the matrix rank. This helps to explain the excellent generalization in gradient-based deep learning, as it converges to a global (or local) minima with the minimum rank.

Although the previous works have proved the existence of GLRL trajectory, it has never been observed in practice due to its impractical requirement of infinitesimal initialization. ZerO's learning trajectory we observed suggests that GLRL not only exists under infinitesimal initialization, but also under initialization around the identity. If a matrix $W$ is initialized as $I$, the low-rank structure is actually inside its residual component: $W_l' = W_l - I$. To be noted, every convolutional kernel conv$(x)$ we measured in Figure 5 can be viewed as the residual component of conv$(x) + I$, where the skip connection is included.

Figure 5 also suggests that the kernels never reach their maximal stable ranks during training under ZerO initialization. This implies that the searching over the space of full-rank weight matrices may be unnecessary, suggesting new avenues towards improved computational efficiency. We hope to explore this direction in future work.

We also observe that ZerO-based networks converge to low-complexity solutions. As shown in both Figure 5 and 6 (left), the convolutional kernels trained by ZerO usually have lower ranks than the kernels trained by random initialization. We further measure model complexity through both network pruning and low-rank approximation.

For network pruning, we use a standard magnitude-based pruning that prunes a portion of weights with the lowest magnitudes in each layer (Frankle & Carbin, 2019). For low-rank approximation, we apply Tucker-2 decomposition over channel dimensions in convolutions to select the most significant components (Kim et al., 2016).

As shown in Figure 6, compared to randomly initialized networks, the sub-networks obtained in trained ZerO-initialized networks achieve 25% higher sparsity or 30% lower (matrix or tensor) rank without sacrificing accuracy. This suggests ZerO encourages the networks to converge to low-complexity solutions, which improves the computational efficiency for inference.

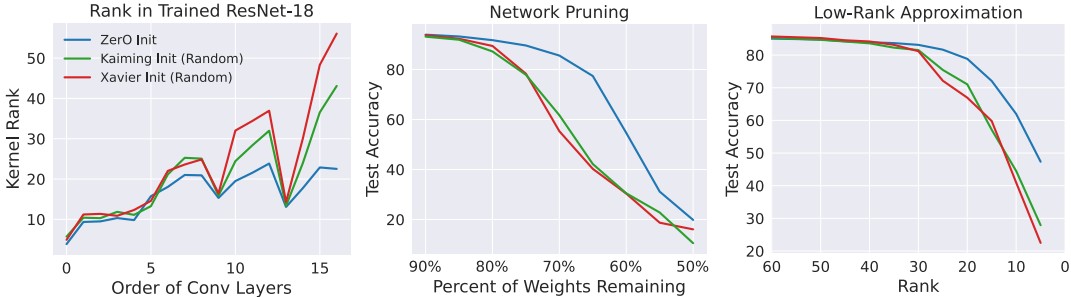

Figure 6: **Left:** comparing kernel rank in ResNet-18 trained by ZerO and Kaiming methods. **Middle:** a magnitude-based network pruning on ResNet-18. **Right:** a Tucker-2 decomposition for a particular convolution with 512 channels in ResNet-18.

## 6 Related Works

To ensure stable training with random weight initialization, previous works such as Glorot & Bengio (2010); He et al. (2015) study the propagation of variance in the forward and backward pass under different activations. Several studies provide a more detailed characterization of the signal propagation with dynamical isometry (Saxe et al., 2014; Pennington et al., 2017; Xiao et al., 2018).

Inspired by the dynamical isometry property, various initialization methods are proposed to increase the convergence speed and stabilize the signal propagation, including Saxe et al. (2014); Bachlechner et al.

(2020); Gehring et al. (2017); Balduzzi et al. (2017). De & Smith (2020); Hoffer et al. (2018) study the reason behind the success of batch normalization (Ioffe & Szegedy, 2015) , and Zhang et al. (2019); De & Smith (2020) propose initialization methods to train residual networks without batch normalization.

As summarized in Table 1, many of these methods can be categorized as identity initialization with random perturbations. However, ZerO eliminates all the randomness using Hadamard and identity transforms. In another related work, Blumenfeld et al. (2020) discusses whether random initialization is needed from the perspective of feature diversity. They propose networks with identical features at initialization, which still require random perturbations to avoid the symmetric problem and improve performance.

Gradient descent biasing models towards low-rank solutions has been well studied in matrix factorization (Arora et al., 2019). Recent works also demonstrate the existence of greedy low-rank learning trajectory induced by gradient descent (Li et al., 2021; Razin et al., 2021; Jacot et al., 2021). However, no prior work demonstrates the greedy low-rank learning trajectory in large-scale applications of deep learning, as most only consider the problems of matrix factorization or applications of shallow neural networks.

## 7 Conclusion

In this work, we propose a simple and fully deterministic initialization called ZerO. Extensive experiments demonstrate that ZerO achieves state-of-the-art performance, suggesting that random weight initialization may not be necessary for initializing deep neural networks. ZerO has many benefits, such as training ultra deep networks (without batch-normalization), exhibiting low-rank learning trajectories that result in low-rank and sparse solutions, and improving training reproducibility.

We believe that ZerO opens up many new possibilities given its various benefits. It can be applied to networks and tasks sensitive to the variances in weight initialization. Its low-rank learning trajectories enable the development of rank-constrained training methods that improve computational efficiency. Finally, the improved training reproducibility can aid model interpretability. We hope our results will inspire other researchers to consider deterministic initialization schemes and to rethink the role of weight initialization in training deep neural networks.

### Acknowledgments

We are grateful to the anonymous reviewers for their helpful comments and NVIDIA for the computational support. Dr. Anandkumar is supported by the Bren chair professorship at Caltech.

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
