# OpenReview forum: "ZerO Initialization: Initializing Neural Networks with only Zeros and Ones"
_TMLR — Accepted by TMLR_

### Review · Reviewer_8dKV · 2022-07-19

**Summary Of Contributions:**

The paper proposes a new deterministic initialization method for residual networks, where all the initial weight values are ones and zeros. They theoretically show that, different from naive identity mapping, their initialization methods can avoid training degeneracy when the network dimension increases. In addition, they empirically show that they can achieve better performance than random initializations on image classification tasks, such as CIFAR-10 and ImageNet. They also show some nice properties of the model trained by their initialization methods, such as low-rank and sparse solutions.


**Broader Impact Concerns:**

I don't have any broader impact concerns.

**Requested Changes:**

1. The main issue to me is that the proposed method is not an initialization method. It can only be applied to a specific network. And it is unknown whether it will work for other networks.

2. We need more experiments to support some of the claims. For example, compared to more initialization methods and with different optimizers or network structures.

3. Captions for Figures 5, and 7 are not informative. What is 1*1*3*5? The kernel size is 1? I cannot tell which figure is for which one in Figure 7.

4. I think making the proposed method work for other types of networks is essential.

**Strengths And Weaknesses:**

Strength:
1. The paper is easy to follow and the proposed method is novel.
2. The empirical results are interesting, where one can achieve comparable results when we have all initial values in ones and zeros.
3. The empirical findings that ZerO initialization leads to low-rank and sparse solutions are interesting.

Weakness:
1. I am concerned about the title of the paper since they only show that their method can work on a modified ResNet-18, which is not a general initialization method for other neural networks. Even for residual networks, they need to add one more skip connection. But the title looks like the proposed method works for all types of networks as an initialization method. Can we apply the proposed method to other models? Moreover, can we apply it to other tasks, especially for the language model?

2. Some statements are overclaimed: 1) The proposed can help train neural networks without BatchNorm. This is not a unique property since other methods can help as well.  2) They claim their method achieves SOTA results. However, they only compared with some basic initialization methods such as Xavier Init and Kaiming Init. However, there're more initialization methods such as FixUp[1], MetaInit[2], GradInit[3], and so on. How is the performance compared to these methods?

3. Why deterministic? I agree that deterministic initialization makes the standard error smaller, but is this property that important? In addition, is this network structure and optimization algorithm dependent? Does this property still hold with other optimizers such as adam or sam?

4. Is there a chance that we can apply the proposed method with other pruning techniques to achieve better empirical results for sparse networks?

6. How to initialize convolutional layers is not clear to me. Do we treat each kernel as a matrix or the whole conv block as a huge matrix? I notice that some of the convolutional layers are initialized to $HI^{\ast}$ and $I$, and some of them are to zero. Is that heuristic? Can we have some ablation study to initialize more convolutional layers to $HI^{\ast}$? I am just curious what will happen if we do not have skip aug or the network does not have residual blocks at all.

[1] Zhang H, Dauphin Y N, Ma T. Fixup initialization: Residual learning without normalization[J]. arXiv preprint arXiv:1901.09321, 2019.
[2] Dauphin Y N, Schoenholz S. MetaInit: Initializing learning by learning to initialize[J]. Advances in Neural Information Processing Systems, 2019, 32.
[3] Zhu C, Ni R, Xu Z, et al. GradInit: Learning to initialize neural networks for stable and efficient training[J]. Advances in Neural Information Processing Systems, 2021, 34: 16410-16422.

---

### Review · Reviewer_ZBoj · 2022-07-27

**Summary Of Contributions:**

In this paper, the authors proposed an initialization scheme for deep neural networks called ZerO. The proposed method use the product of identity and Hadamard matrix as initial weight for training. Theoretical analysis show that compared with identity initialization, the model trained with ZerO initialization does not have rank constrained by input dimension. Experiments with ResNet on image classification tasks show the ZerO initialization achieves similar results as other initialization methods.

**Broader Impact Concerns:**

To my knowledge this work does not pose any negative societal impact.

**Requested Changes:**

I'd like to see more discussion on the motivation behind the method design, e.g. what are the considerations and what are the gains.

How does the ZerO perform on vanilla ResNet (without AugSkip)? It seems to be missing from Table 2.

There's only empirical study on ResNet for image classification tasks, more empirical study on more types of DNNs and problems (e.x. Transformers on NLP tasks) is needed to fully understand the effect of ZerO initialization.


**Strengths And Weaknesses:**

Although there are a large amount of study on the initialization of DNNs, I do think this work provide an interesting option to approach the problem. However, I have several concerns on the novelty and significance of the work.
Firstly, I think there's lack of motivation behind having a fully deterministic initialization method as opposed to a random initial point. The final model is still not deterministic with the stochastic mini-batch training.

Also, it is not clear why the authors have chosen the Hadamard matrix. I might be missing something but after reading the paper I still don't get how Hadamard matrix is different from any other orthogonal or simply full-rank matrix in this case?
In addition, the singular value of Hadamard matrix is linear to $\sqrt{n}$, wouldn't this lead to exploding gradients at the beginning of training?

---

### Review · Reviewer_qqPa · 2022-07-29

**Summary Of Contributions:**

This paper proposed a deterministic initialization method for ResNet where weights are initialized with zero, identity or a Hadamard matrix depending on the type/location of the layer. The proposed ZerO initialization also used a variant of ResNet with augmented skip connections. Experiments performed on CIFAR-10 and ImageNet suggest ZerO init can be comparable with Kaiming and Xavier initialization while being more stable as a deterministic initialization. ZerO init also significantly outperforms Fixup, Rezero and Kaiming init when evaluated on CIFAR-10 over 15 epochs for ResNet with more than 150 layers.


**Broader Impact Concerns:**

No concerns.

**Requested Changes:**

Is it possible to provide more ablation study/baseline methods in table 2? ZerO is only compared to Kaiming and Xavier init now. If I understand correctly, ZerO shares a lot of common parts such as the zero and identity initialization with Rezero and ConstNet, so it can be beneficial to compare these methods to show the effect of the proposed change: use Hadamard transformation for initialization.

Could the authors discuss the motivation of the AugSkip architecture changes in Figure 4, and whether ZerO works for common ResNet structure?

I have some concerns about Figure 6 for the evaluation on CIFAR-10 over 15 epochs for ResNet with various depths. How is the number of “epoch 15” chosen? Fixup [Zhang 2019] shows a figure for epoch 1 and the accuracy drop is very small up to depth 1000. Both the original ResNet [He 2015] and GradInit [Zhu 2021] train ResNet-1202 and achieve comparable or even better performance than shallower ResNets.

I would encourage the authors to discuss [Dauphin 2019 MetaInit: Initializing learning by learning to initialize] and [Zhu 2021 GradInit: Learning to Initialize Neural Networks for Stable and Efficient Training]. Additionally and optionally, [Gilmer 2021 A Loss Curvature Perspective on Training Instability in Deep Learning] suggests successful initialization (Fixup, MetaInit, GradInit, Warmup) can change the loss landscape. It might be good to study the effect of the proposed ZerO initialization.


**Strengths And Weaknesses:**

Strength
+ Initialization is an important topic for neural networks training.
+ A deterministic method can benefit reproducibility.
+ The performance looks good on CIFAR-10 over 15 epochs for ResNet with more than 150 layers.
+ The paper is generally well written and easy to follow.

Weakness
- The writing and experiments also make it not so easy to compare the proposed method ZerO with existing methods Rezero [Bachlechner 2020] and ConstNet [Blumenfeld 2020]. If I understand correctly, there are two changes: (1) use Hadamard transformation instead of random initialization for some layers (2) the additional residual connection (AugSkip).
- The AugSkip architecture change looks ad-hoc. Could the authors provide more theoretical support or insights? Does ZerO work for common ResNet structures?
- The experiments are only on ResNet for CIFAR-10 and ImageNet, and can be improved for clarity with more ablation study.
- Some related works are not discussed. See below for details

---

### Decision · Action_Editors · 2022-09-08

**Recommendation:** Accept as is

**Comment:**

This paper proposes a novel initialization scheme for deep residual networks called ZerO. The proposed method relies on the use of Hadamard and zero matrices and they show theoretically that this scheme avoids training degeneracy. The reviewers raised issues with the lack of baselines, which were addressed by the authors in the rebuttal phase with Table 3. There were also concerns about the generality of this method since it applied only to modified resnets. This concern was also addressed in revisions by the authors. Experiments on Imagenet and WikiText-2 show that this method can be effective to train very deep networks.